# Evaluation of Empirical Reference Evapotranspiration Models Using Compromise Programming: A Case Study of Peninsular Malaysia

**Mohd Khairul Idlan Muhammad [1], Mohamed Salem Nashwan [1,2], Shamsuddin Shahid [1], Tarmizi bin Ismail [1], Young Hoon Song [3] and Eun-Sung Chung [3,\*]**

[1] School of Civil Engineering, Faculty of Engineering, Universiti Teknologi Malaysia (UTM), Johor Bahru 81310, Malaysia
[2] Faculty of Engineering and Technology, Arab Academy for Science, Technology and Maritime Transport (AASTMT), Elhorria 2033, Cairo, Egypt
[3] Faculty of Civil Engineering, Seoul National University of Science and Technology, 232 Gongneung-ro, Nowon-gu, Seoul 01811, Korea
\* Correspondence: eschung@seoultech.ac.kr; Tel.: +82-2-970-9017

**Abstract:** Selection of appropriate empirical reference evapotranspiration ($ET_o$) estimation models is very important for the management of agriculture, water resources, and environment. Statistical metrics generally used for performance assessment of empirical $ET_o$ models, on a station level, often give contradictory results, which make the ranking of methods a challenging task. Besides, the ranking of $ET_o$ estimation methods for a given study area based on the rank at different stations is also a difficult task. Compromise programming and group decision-making methods have been proposed in this study for the ranking of 31 empirical $ET_o$ models for Peninsular Malaysia based on four standard statistical metrics. The result revealed the Penman-Monteith as the most suitable method of estimation of $ET_o$, followed by radiation-based Priestley and Taylor and the mass transfer-based Dalton and Meyer methods. Among the temperature-based methods, Ivanov was found the best. The methodology suggested in this study can be adopted in any other region for an easy but robust evaluation of empirical $ET_o$ models.

**Keywords:** evapotranspiration; Malaysia; empirical model; radiation; temperature

## 1. Introduction

Evapotranspiration (ET), the process of water release to the atmosphere, plays a crucial role in irrigation management [1], water balance estimation [2], surface runoff estimation [3], groundwater level prediction [4], water stress assessment [5], reservoir management [6], daily flux modelling [7], and climate change impact assessment [8]. It determines the crop irrigation requirement and thus, irrigation management, the introduction of new crop, or crop scheduling to adapt to climate change [9–11]. It is a major component that defines surface runoff and therefore, important for designing drainage and hydraulic structure [12,13]. In addition, it is the major component that determines the ecological or environmental water demand and thus, assessment of environmental sustainability or ecological balance [14]. It provides an assessment of water release from surface water bodies and reservoirs to the atmosphere and therefore, operation and management of water resources [15,16]. Hence, ET is considered as a vital component for any hydrological and climatic study [17]. Atmospheric water is an important driving factor of precipitation [18]. It has a significant effect on the retention of solar radiation and thus, controlling the air temperature of a region [19]. Therefore, the importance of the assessment of ET becomes more crucial in the context of climate change.

A most accepted method of ET estimation is to measure the reference evapotranspiration ($ET_o$) [20]. In-situ measurement of $ET_o$ is expensive and time-consuming, and subject to significant uncertainties. Because of the limitation of in-situ measurements of $ET_o$, many empirical models have been developed to estimate $ET_o$ in the last 70 years, since the introduction of the Thornthwaite method in 1944 [21]. The $ET_o$ depends on atmospheric energy balance and release of water to the atmosphere from vegetation [22,23]. Therefore, the $ET_o$ estimation methods are categorized according to the meteorological parameters they use. The $ET_o$ method has been divided into different categories in different studies. Most widely, it is classified into four groups: (i) Water balance/mass transfer; (ii) radiation; (iii) temperature; and (iv) combination of the aforementioned. Each method has its own perspectives, concepts, and often developed for a particular climatic region. Few of them are developed through modification of other established methods. However, the main challenge in the estimation of $ET_o$ is the skill of the method used [15,24]. Most of the ET estimation methods are developed for a particular region with a specific viewpoint, and therefore, they are often found inefficient in estimating $ET_o$ in other climatic zones. However, some methods are developed without focusing on any climatic region and have been found applicable over a wide range of climate. The major challenge arises in the selection of the best model for an area with the least error compared to in-situ measurements.

ET is a crucial element in defining the water budget and physical processes in tropics. The condensation of the vast volume of water vapor in the tropical region leads to the release of latent heat energy to the atmosphere, which is very important for climatology in the region. Tropical regions, particularly the Southeast Asian tropical region, are rich in biodiversity. This rich biodiversity is promoted by high rainfall and high ET, among other factors. Changes in ET can have a severe impact on tropical biodiversity, and therefore, monitoring of ET is very important for the region. It is particularly crucial for Peninsular Malaysia where about 60% of its land is covered with forest with dense biodiversity.

A large number of studies have been conducted to select the most suitable $ET_o$ model in different parts of the globe [15,20,25–28], including Peninsular Malaysia [13,29–32]. Ali et al. [30] and Ali et al. [31] found a strong agreement of the monthly average of class A pan evaporation with the FAO Penman-Monteith [33] estimation for the Muda irrigation project, the largest paddy field in Malaysia, in the north of the Peninsula. Tukimat et al. [13] compared a number of temperature- and radiation-based methods with the FAO Penman-Monteith model to estimate $ET_o$ in the Muda irrigation project, and found that the radiation-based models give better estimates of $ET_o$. Lee et al. [29] compared the pan evaporation with the estimates of eight empirical models and found a good agreement between pan evaporation estimates of $ET_o$ with the estimates of the FAO Penman-Monteith and FAO Blaney-Criddle [34] models in the west coast of the Peninsular. Muniandy et al. [32] compared the pan evaporation estimates with 26 empirical model estimations at a station located in the south of the Peninsular, and reported that the mass transfer-based Penman model can provide better estimates of $ET_o$ compared to other methods.

Different statistics have been used in previous studies for the assessment of the performance of $ET_o$, which include root mean square error (RMSE), mean absolute error (MAE), Nash-Sutcliff efficiency (NSE), bias ratio, etc. Selection of $ET_o$ method based on a single statistic like RMSE or NSE is questionable, as these statistics can be used for the estimation of a particular property only. For example, RMSE provides a measure of the mean distance between two time series, while correlation provides how two time series follow each other in their variation. The correlation coefficient ($R^2$) can be excellent even if the distance between the two series is high, while RMSE can be much less even if one time series fails to follow the variation of another series. Thus, a number of statistical metrics are generally used for the assessment of the performance of different $ET_o$ methods [13,15,25–29]. However, the major problem with using a number of statistical metrics is that different metrics often provide contradictory results [35–37]. For example, a model may show good agreement in terms of RMSE, but a worse measure in terms of $R^2$. Thus, it often becomes challenging to make a decision based on different statistics.

Compromise programming (CP) [38] can be used to find the most suitable solution through judicious compromising of different objectives, among which many may be conflicting. CP attempts to identify a solution where all the considered objectives achieve the most suitable value [39,40]. CP has been found more efficient compared to conventional multi-criteria decision analysis (MCDA) methods in finding the most suitable solution [38,40–44].

The ranking of the $ET_o$ estimation method at a single station based on the ability to replicate the observed ET can be done using CP and a matrix of statistical indices. However, it is often required to suggest the best $ET_o$ model for a region based on the performance at different stations over the region. Ranking of $ET_o$ based on the performance at multiple stations becomes challenging, as different $ET_o$ models often show different ranks at different stations. Group decision making (GDM) can be employed for such cases where the $ET_o$ model is given a position based on the frequency of the rank obtained at different stations.

The objective of the present study is to use CP for the ranking of empirical $ET_o$ models for Peninsular Malaysia. Four statistical metrics were used for the assessment of the performance of 31 $ET_o$ models at 10 locations distributed over the Peninsula. CP was used for the ranking the empirical models at each of the 10 stations. Finally, an information aggregation approach was used for the ranking of the empirical models for the entire Peninsular Malaysia based on the results obtained at the different stations. This is the first approach of the ranking of empirical $ET_o$ models based on CP and the information aggregation approach. The method proposed in this study can be used for the ranking of empirical models in a prudent way.

## 2. Study Area and Data

### 2.1. Geography and Climate of Peninsular Malaysia

Situated along the tropics, Peninsular Malaysia covers an area of 130,598 km$^2$ (Figure 1). Undulating mountains in the middle and relative flat coast on all the three sides (east, west, and south) are the major topographic features of the peninsula. About 60% of the land is covered by forest. The year-round rainfall, high uniform temperature, and high humidity are the major characteristics of the climate of Peninsular Malaysia. The climate is more or less homogeneous throughout the Peninsula [45,46]. Due to its geographical location, the weather in the region is influenced by both the northeast and the southwest monsoon, and thus experiences a significant amount of rainfall even in the driest month. The annual average rainfall in Peninsular Malaysia varies between 1950 and 4000 mm [47]. The number of rainfall days ranges between 150 and 200. Weather is always hot due to its location in the tropics, and humid due to high rainfall. The mean temperature in the peninsula varies between 23 °C in the central highlands and 32 °C in the coastal region [46,48]. Seasonal variation of mean temperature is always less than 2.0 °C from the mean temperature of 27 °C. Being located in the equator, the study area receives long daylight hours (about 12 h) throughout the year and, thus, sufficient solar radiation. The wind in peninsular Malaysia is mostly light (0.9 to 2.3 m/s). Sunshine hours and temperature have an important role in $ET_o$ in the study area. The $ET_o$ is lower in the rainy season due to lower sunshine hours. Furthermore, it is lower in central mountainous areas (2.5 mm/day) due to relatively higher humidity compared to the coastal region (4–5 mm/day), where the humidity is less.

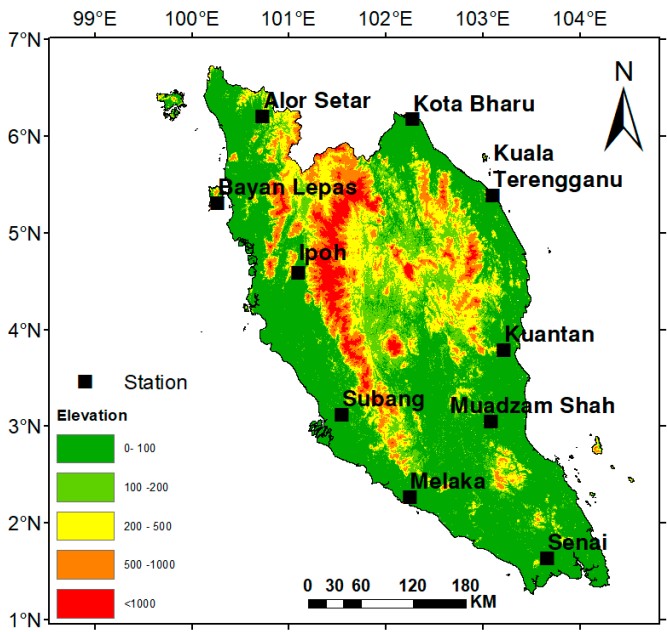

**Figure 1.** The geographical location of Peninsular Malaysia and the selected meteorological station used in this study. Description of the stations is given in Table 1.

## 2.2. Data and Sources of Information

The observed daily meteorological data of temperature (mean, maximum, and minimum), relative humidity, solar radiation, wind speed, and pan evaporation from 10 meteorological stations fairly distributed over Peninsular Malaysia were collected from the Malaysian Meteorological Department. The locations of the meteorological stations are shown in Figure 1. The summary of the different climatic variables used in the present study is given in Table 1.

**Table 1.** Descriptive statistics of the meteorological stations used in the present study.

| Station Name | Station No | Elev. (m) | Time Period | Tmax (°C) | Tmin (°C) | Tmean (°C) | RH (%) | u (m/s) | $R_s$ (MJ/m$^2$) | ETpan (mm/day) |
|---|---|---|---|---|---|---|---|---|---|---|
| Alor Star | 48603 | 3.9 | 1985–2014 | 32.4 | 23.4 | 27.1 | 81.2 | 1.5 | 18.5 | 3.4 |
| Bayan Lepas | 48601 | 2.5 | 1985–2014 | 31.4 | 23.9 | 27.2 | 80.4 | 1.8 | 17.9 | 2.9 |
| Kota Bharu | 48615 | 4.4 | 2000–2014 | 31.2 | 23.5 | 26.9 | 80.6 | 2.3 | 18.9 | 3.2 |
| Ipoh | 48625 | 40.1 | 1985–2014 | 33.0 | 23.3 | 27.0 | 81.3 | 1.4 | 17.7 | 3.1 |
| Kuala Terengganu | 48618 | 5.2 | 2005–2008 | 31.4 | 23.8 | 27.0 | 82.4 | 1.9 | 17.5 | 3.3 |
| Subang | 48647 | 16.6 | 1985–2014 | 32.4 | 23.2 | 26.9 | 79.6 | 1.5 | 16.5 | 3.2 |
| Kuantan | 48657 | 15.2 | 1999–2014 | 31.7 | 22.9 | 26.2 | 84.1 | 1.7 | 17.0 | 2.9 |
| Muadzam Shah | 48649 | 33.3 | 1985–2014 | 32.1 | 22.7 | 26.3 | 84.7 | 0.9 | 16.3 | 2.5 |
| Melaka | 48665 | 9.0 | 1999–2010 | 31.9 | 23.2 | 26.8 | 80.1 | 1.7 | 17.0 | 3.3 |
| Senai | 48679 | 37.8 | 1985–2010 | 31.8 | 22.5 | 26.0 | 85.7 | 1.4 | 15.1 | 2.7 |

RH is the relative humidity; u is the wind speed; $R_s$ is the solar radiation; and ETpan is the pan evaporation.

Pan evaporation is an indirect and less expensive method of estimation of ET and therefore, it is most widely used for estimation of ET. The pan evaporation data is multiplied by the pan coefficient to get the $ET_o$. The pan coefficient value varies between 0.35 and 0.85, depending on the nature of the evaporating surfaces (land use), altitude, average humidity, and average wind speed of the site [49]. Considering the existing setup, the Department of Irrigation and Drainage of Malaysia [50] suggested a pan coefficient of 0.75 for the estimation of $ET_o$ from pan evaporation in Malaysia. Therefore, the observed $ET_o$ was calculated by multiplying the pan evaporation data by the pan coefficient of 0.75.

## 3. Methodology

The performance of different empirical $ET_o$ models was assessed and ranked by comparing their estimations with the in-situ data. The methodology adopted in this study is summarized below.

1. $ET_o$ was estimated by the empirical models using the metrological variables.
2. Four statistical metrics were used to estimate the capability of different empirical $ET_o$ models to estimate different properties of observed $ET_o$ at each station.
3. CP was used to integrate the results of statistical metrics and rank the $ET_o$ models at each station.
4. GDM, an information accumulation method, was deployed to rank the empirical models for the entire Peninsula.

### 3.1. Empirical $ET_o$ Models

In this study, 31 empirical $ET_o$ models were evaluated by comparing their estimates with the pan evaporation data. They were selected based on their applicability worldwide and the availability of required input data. The empirical models were classified into four groups based on the input parameters. Out of 31 models, 10 are temperature-based, 10 are radiation-based, 10 are mass transfer-based models, and one is a combination model. The $ET_o$ was calculated using the meteorological input at each station location without any calibration. Table 2 lists the input parameters and the equation of each of the 31 empirical models.

**Table 2.** List of the empirical reference evapotranspiration ($ET_o$) models evaluated in this study, along with their input parameters and equations. They are classed into four groups: Temperature-based, radiation-based, mass transfer-based, and combination.

| No | Model | Input Parameter | Equation |
|---|---|---|---|
| | | **Temperature-based** | |
| 1 | Ivanov [51] | $T_{mean}$, RH | $ET_o = 0.00006(25 + T_{mean})^2(100 - RH)$ |
| 2 | Hamon [52] | $T_{mean}$ | $ET_o = 0.1651 L_d \, \text{RHOSAT} \times \text{KPEC}$ <br> $\text{RHOSAT} = \frac{216.7\text{ESAT}}{T_{mean}+273.3}$ <br> $\text{ESAT} = 6.108 \exp\left(\frac{17.269 \times T_{mean}}{T_{mean}+273.3}\right)$ |
| 3 | Papadakis [53] | $T_{mean}$, RH | $ET_o = 2.5(e_{ma} - e_a)$ |
| 4 | Schendel [54] | $T_{mean}$, RH | $ET_o = 16\left(\frac{T_{mean}}{RH}\right)$ |
| 5 | FAO Blaney-Criddle [34] | $T_{mean}$ | $ET_o = p(0.46 T_{mean} + 8.13)$ |
| 6 | Linacre [55] | $T_{mean}$ | $ET_o = \frac{\frac{700(T_{mean} \pm 0.006z)}{100-L} + 15(T_{mean} - T_d)}{80 - T_{mean}}$ |
| 7 | Kharrufa [56] | $T_{mean}$ | $ET_o = 0.34 p T_{mean}^{1.30}$ |
| 8 | Hargreaves et al. [57] | $T_{mean}$, $T_{min}$, $T_{max}$, $R_a$ | $ET_o = \left(0.0023\frac{R_a}{2.45}\right)TD^{0.5}(T_{mean} + 17.8)$ |
| 9 | Trajkovic [58] | $T_{mean}$, $T_{min}$, $T_{max}$, $R_a$ | $ET_o = (0.0023 R_a)TD^{0.424}(T_{mean} + 17.8)$ |
| 10 | Ravazzani et al. [59] | $T_{mean}$, $T_{min}$, $T_{max}$, $R_a$ | $ET_o =$ <br> $(0.817 + 0.00022z)(0.0023 R_a)\left(TD^{0.5}\right)(T_{mean} + 17.8)$ |
| | | **Radiation-based** | |
| 11 | Makkink [60] | $T_{mean}$, $R_s$ | $ET_o = 0.61\left(\frac{\Delta}{\Delta+\gamma}\right)\frac{R_s}{58.5} - 0.12$ |
| 12 | Turc [61] | $T_{mean}$, $R_s$, RH | $ET_o = 0.013\left(\frac{T_{mean}}{T_{mean}+15}\right)(R_s + 50)$ |
| 13 | Jensen et al. [62] | $T_{mean}$, $R_s$ | $ET_o = \left(\frac{R_s}{\lambda}\right)(0.025 T_{mean} + 0.08)$ |
| 14 | Priestley et al. [63] | $T_{mean}$, $R_s$, RH | $ET_o = \alpha\left(\frac{\Delta}{\Delta+\gamma}\right)\frac{R_n}{\lambda}$ |
| 15 | McGuinness et al. [64] | $T_{mean}$, $R_s$ | $ET_o = (0.0082 T_{mean} - 0.19)\frac{R_s}{1500}(2.54)$ |
| 16 | Caprio [65] | $T_{mean}$, $R_s$ | $ET_o = \left(\frac{6.1}{10^6}\right)R_s(1.8 T_{mean} + 1.0)$ |
| 17 | Jones et al. [66] | $T_{min}$, $T_{max}$, $R_s$ | $ET_o = \alpha_1\left(3.87 \times 10^{-3}\right)(R_s(0.6 T_{max} + 0.4 T_{min} + 29))$ |
| 18 | Abtew [67] | $T_{mean}$, $R_s$ | $ET_o = 0.53\left(\frac{R_s}{\lambda}\right)$ |
| 19 | Irmak et al. [12] -Rs | $T_{mean}$, $R_s$ | $ET_o = -0.611 + 0.149 R_s + 0.079 T_{mean}$ |
| 20 | Irmak et al. [12] -Rn | $T_{mean}$, $R_s$, RH | $ET_o = 0.489 + 0.289 R_n + 0.023 T_{mean}$ |

**Table 2.** *Cont.*

| | | | |
|---|---|---|---|
| **Mass transfer-based** | | | |
| 21 | Dalton [68] | $T_{mean}$, RH, u | $ET_o = (0.3648 + 0.07223(u))(e_s - e_a)$ |
| 22 | Trabert [69] | $T_{mean}$, RH, u | $ET_o = (0.3075) \sqrt{u}(e_s - e_a)$ |
| 23 | Meyer [70] | $T_{mean}$, RH, u | $ET_o = (0.375 + 0.05026(u))(e_s - e_a)$ |
| 24 | Rohwer [71] | $T_{mean}$, RH, u | $ET_o = (3.3 + 0.891(u))(e_s - e_a)$ |
| 25 | Penman [72] | $T_{mean}$, RH, u | $ET_o = (2.625 + 0.000479/u)(e_s - e_a)$ |
| 26 | Albrecht [73] | $T_{mean}$, RH, u | $ET_o = (0.1005 + 0.297(u))(e_s - e_a)$ |
| 27 | Brockamp et al. [74] | $T_{mean}$, RH, u | $ET_o = 0.543(u^{0.456})(e_s - e_a)$ |
| 28 | WMO [75] | $T_{mean}$, RH, u | $ET_o = (0.1298 + 0.0934(u))(e_s - e_a)$ |
| 29 | Mahringer [76] | $T_{mean}$, RH, u | $ET_o = (0.15072) \sqrt{3.6u}(e_s - e_a)$ |
| 30 | Szasz [77] | $T_{mean}$, RH, u | $ET_o = 0.00536(T_{mean} + 21)^2(1 + RH)^{2/3} f(u)$ <br> $f(u) = (0.0519u) + 0.905$ |
| **Combination-based** | | | |
| 31 | FAO Penman-Monteith [33] | $T_{mean}$, $R_s$, RH, u, $e_s$ | $ET_o = \dfrac{0.408(R_n - G) + \gamma \frac{900}{T_{mean}+273} u(e_s - e_a)}{\Delta + \gamma(1 + 0.34u)}$ |

$ET_o$ is the evapotranspiration in mm/day in all equations except the Ritchie and McGuinness and Bordne models, where $ET_o$ is in cm/day. $R_n$ is the net radiation (MJ/m$^2$/day). $G$ is the soil heat flux (MJ/m$^2$/day). $R_a$ is the extraterrestrial radiation (MJ/m$^2$/day). $\Gamma$ is the psychrometric constant (kPa/°C). $e_s$ is the saturation vapor pressure (hPa). $e_a$ is the actual vapor pressure (hPa). $e_s$ and $e_a$ are in hPa in all equations except the Papadakis, Rohwer, Penman, and FAO Penman-Monteith models, where $e_s$ and $e_a$ are in kPa. $\Delta$ is the slope of the saturation vapor pressure–temperature curve (kPa/°C). $\lambda$ is the latent heat of evaporation (MJ/kg). $T_{mean}$ is the average daily air temperature (°C). $T_{mean}$ is in °C in all equations except the McGuinness and Bordne model, where $T_{mean}$ is in °F. $u$ is the mean daily wind speed at 2 m (m/s). $f(u)$ is a function of wind speed. $Z$ is the elevation (m). L is local latitude (degrees). $T_d$ is the dew point temperature (°C). $T_{min}$ is the minimum air temperature (°C). $T_{max}$ is the maximum air temperature (°C). TD is the maximum and minimum temperature difference (°C). RH is the average relative humidity (%). $Rs$ is the solar radiation. $Rs$ is in MJ/m$^2$/day in all equations except the Turc, Makkink, Ritchie and McGuinness, and Bordne models, where $Rs$ is in Cal/m$^2$ day, and the Caprio model, where $Rs$ is in kJ/m$^2$ day. $e_{ma}$ is the saturation vapor pressure at the monthly mean daily maximum temperature (kPa). $p$ is the mean annual percentage of daytime hours for different latitudes that can be obtained from Doorenbos et al. [34]. $p$ is expressed as constant (0.274) in Muniandy et al. [32]. $L_d$ is the daytime length in multiples of 12 h. RHOSAT is saturated vapor density (g/m$^3$). ESAT is the saturated vapor pressure (mbar). KPEC is the calibration coefficient (1.2). $\alpha$ is a constant (1.26). $\alpha_1$ is a constant (1.1).

*3.2. Statistical Indices*

Four statistical metrics were used to measure the capability of each empirical model in estimating the observed $ET_o$ at each gauge location. They were the normalized root mean square error (NRMSE), percentage of bias (%BIAS), modified index of agreement (md), and Kling-Gupta efficiency (KGE). The NRMSE is a measure of accuracy as it calculates the magnitudes of the errors in modeled data [78]. The %BIAS quantifies the tendency of $ET_o$ estimation by empirical models to under or over-estimate the observed data [36]. The md summarizes the additive and proportional differences in the observed and modeled $ET_o$ means and variances. The KGE integrates linear correlation (*r*), bias ratio (β), and variability (γ) of observed and modeled data [35,79]. Table 3 presents each metric equation, range, and optimum value.

**Table 3.** The metric equations, range, and optimum value.

| Metric Equation | | Range | Optimum Value |
|---|---|---|---|
| $NRMSE = \dfrac{\left[\frac{1}{n}\sum_{i=1}^{n}(ET_{0m,i}-ET_{0obs,i})^2\right]^{\frac{1}{2}}}{\frac{1}{n}\sum_{i=1}^{n}(ET_{0m,i})}$ | (1) | 0 to $+\infty$ | 0 |
| $\%BIAS = 100 \times \left[\dfrac{\sum_{i=1}^{n}(ET_{0m,i}-ET_{0obs,i})}{\sum_{i=1}^{n}ET_{0m,i}}\right]$ | (2) | $-\infty$ to $+\infty$ | 0 |
| $md = 1 - \dfrac{\sum_{i=1}^{n}(ET_{0obs}-ET_{0m})^j}{\sum_{i=1}^{n}\left(\left|ET_{0m}-\overline{ET_{0obs}}\right|+\left|ET_{0obs}-\overline{ET_{0obs}}\right|\right)^j}$ | (3) | 0 to 1 | 1 |
| $KGE = 1 - \sqrt{(r-1)^2 + (\beta-1)^2 + (\gamma-1)^2}$ | (4) | $-\infty$ to 1 | 1 |

$ET_{0m,i}$ and $ET_{0obs,i}$ are the $i$-th modeled and observed $ET_0$ data; $n$ is the number of observations; $j$ represents an arbitrary positive power; $r$ is the Pearson correlation; $\beta$ is the bias ratio; and $\gamma$ represents the variability of observed and modeled data.

### 3.3. Compromise Programming

Compromise programming (CP) was used to integrate the results of the statistical metrics described above to enable selection of the most accurate empirical $ET_o$ model. CP ranks the empirical methods based on the distance of each method from an ideal value for the set [42,80]. The CP index (CPI) can be calculated as follows.

$$CPI = \left[\sum_{i=1}^{n}\left|x_i - x_i^*\right|^p\right]^{1/p} \tag{5}$$

where $i$ represents the result of a statistical metric; $x_i$ is the normalized value of metric $i$ of the empirical model; and $x_i^*$ is the normalized ideal value of the metric $i$. The parameter $p$ is used to measure the distance of a solution from an ideal point. The $p$ can have a value between 1 and $\infty$. However, 1, 2, and $\infty$ are most commonly used in CP [81,82]. Therefore, these values are used in this study to estimate the CPI. The differences between the observed value of the metrics and $x_i^*$ are directly proportional to their magnitude when $p = 1$. The higher differences have greater influence in the case of $p = 2$. When $p = \infty$, the minimum values of the maximum differences are used for the estimation of the CPI. Details of the method can be found in [37,80].

In this study, we considered equal importance of all the $ET_o$ estimation models and therefore, the weight parameter of the CP method proposed by Zeleny [38] is not considered. The CPI value ranges between zero and positive infinity, where zero is the most preferable value.

### 3.4. Ranking the Empirical $ET_o$ Models

The ranking of empirical models in estimating observed $ET_o$ from several stations was a challenging task. This was due to the fact that a model may show various degrees of accuracies at different locations. To overcome this challenge, information aggregation methods, such as mean ranking, majority of ranks, and frequency of occurrence, were useful [42,83]. They integrate information from different sources to help in the decision-making process [84]. In this study, empirical models were ranked using GDM. The ranking procedure is outlined below.

1. The empirical models were ranked at station level using their CPI (from 1 to 31, the lowest CPI was ranked 1st).
2. The frequency of occurrence ($F$) of each model of getting a certain rank at all stations was calculated through a $31 \times 31$ matrix.
3. The rank positions were given weight as the inverse of the rank $\left(w_r = rank^{-1}\right)$.
4. The frequency of occurrence of a model at a certain rank, obtained in Step 2, was multiplied by the weight of the rank, obtained in Step 3.
5. The overall score of each $ET_o$ model ($W_m$) was estimated by adding the output of Step 4 as presented in Equation (6).

6. The empirical models were ranked according to the calculated overall weight, where the highest weighted model was ranked top (1st position).

$$W_m = F_1(w_{r1}) + F_2(w_{r2}) + F_3(w_{r3}) + \ldots + F_{31}(w_{r31}) \tag{6}$$

## 4. Results

The $ET_o$ was estimated using all the 31 empirical models at each station, using the meteorological variables. Figure 2 shows heat-scatter plots of observed $ET_o$ against each empirical model estimation of $ET_o$ for all the stations. It can be seen from Figure 2a–t, all the temperature- and radiation-based models tend to overestimate the observed $ET_o$ except for the Ivanov and Makkink model. The overestimation was generally lower by the radiation-based models (Figure 2k–t) than the temperate-based models, which indicates that the overestimation may be due to the exclusion of other factors influencing $ET_o$ in the study area. The Ritchie model was found to heavily overestimate the observed $ET_o$, as seen in Figure 2q. Overall, the mass transfer-based models' estimations (Figure 2u–ad) were found to be more aligned to the 1:1 diagonal line than the temperature- and radiation-based methods. The Penman, WMO, and Mahringer models underestimated the observations. The FAO Penman-Monteith model estimations of $ET_o$ were aligned with the 1:1 line (Figure 2ae).

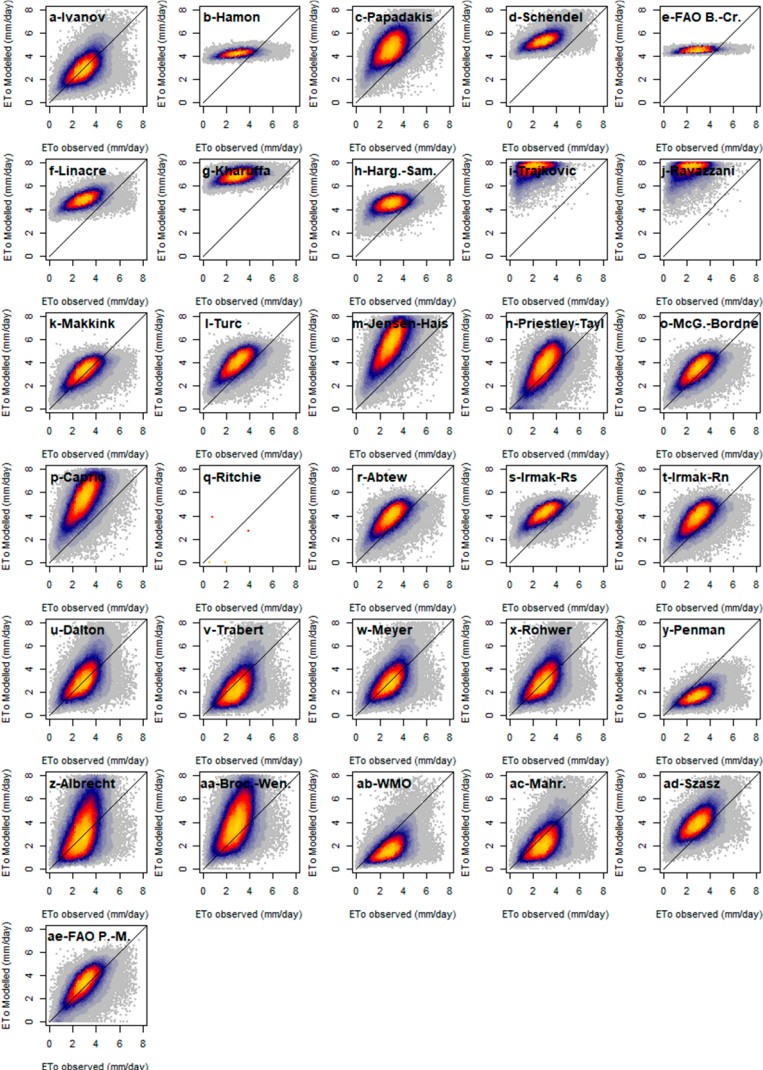

**Figure 2.** Heat-scatter plots of the observed $ET_o$ against the estimation of empirical models (**a**–**ae**).



### 4.1. Evaluation Using Statistical Metrics

The statistical metrics obtained at all the 10 station locations by comparing the observed $ET_o$ with the different empirical model estimations are presented as box plots in Figure 3. The blue, green, gold, and pink box plots represent the temperature-, radiation-, mass transfer-, and combination-based methods, respectively. The red vertical lines represent the optimum value of each metric. Overall, most of the temperature-based methods were found to be poor at estimating the $ET_o$. Among the temperature-based methods, the Ivanov model was found preferable, which had a median NRMSE of 108.8, median %BIAS of 0.70%, median md of 0.51, and median KGE of 0.44.

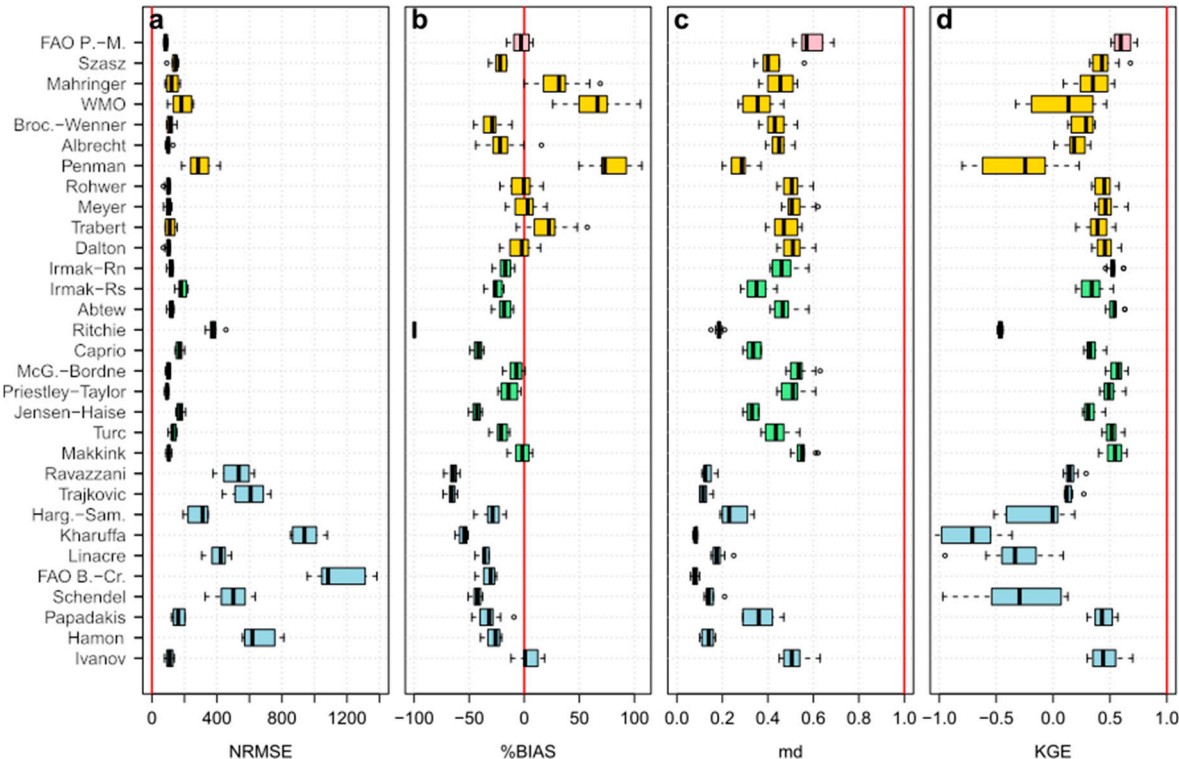

**Figure 3.** Box plot of (**a**) normalized root mean square error (NRMSE), (**b**) percentage of bias (%BIAS), (**c**) modified index of agreement (md), and (**d**) Kling-Gupta efficiency (KGE) obtained for different empirical models in estimating the $ET_o$. The blue, green, gold, and pink box plots represent the temperature-based, radiation-based, mass transfer-based, and combination-based models, respectively. The red vertical lines represent the optimum value of each metric.

The performance of radiation-based methods was found superior to temperature-based models in estimating $ET_o$. The Makkink model was the best performing model among them, and the Ritchie was the worst. As shown in Figure 3, Makkink had the lowest median %BIAS (−1.80%), and highest median md (0.55). However, the Priestley and Taylor model had a slightly better median NRMSE (94.35) than Makkink (102.10). The McGuinness and Bordne model had a better median KGE (0.57) than Makkink (0.55).

Among the mass transfer-based models, the Rohwer and Meyer's methods performed best. Rohwer had median NRMSE, %BIAS, md, and KGE of 104.25, −0.40%, 0.51, and 0.45, respectively. The Meyer model had median NRMSE, %BIAS, md, and KGE of 103.45, 3.10%, 0.51, and 0.45, respectively. The FAO Penman-Monteith had a median NRMSE of 85, %BIAS of −2.90%, md of 0.57, and KGE of 0.60.

The FAO Penman-Monteith model had the lowest NRMSE median, and the highest md and KGE medians. However, the Rohwer model had a lower %BIAS median than the FAO Penman-Monteith model. The Rowher, Meyer, and Makkink models had similar NRMSE, but Rowher had the lowest

%BIAS, and Makkink had the highest md and KGE medians. Therefore, it is important to use CP to integrate the results of the statistical metrics to make a concrete evaluation decision.

*4.2. Compromise Programming*

CP was employed to integrate the statistical metrics and rank the empirical models based on their capability in estimating the observed $ET_o$ in Peninsular Malaysia. It was used to measure the distance of each empirical model from an ideal point at each station separately. As an example, the ideal results obtained at Kuantan station were the lowest NRMSE (93.10), the %BIAS nearest to zero (3.70) and, the highest md and KGE (0.58, for both). The CPI was calculated for each model through the summation of the subtraction of each metric from the ideal value. The following equation presents an example of the CPI calculation of the Ivanov model at Kuantan.

$$CPI_{KU,Ivanov} = |136.60 - 93.10| + |12.30 - 3.70| + |0.50 - 0.58| + |0.30 - 0.58| = 52.46 \qquad (7)$$

The same procedure was used to calculate the CPI of the remaining models. Figure 4 shows a level plot of the CPI for each empirical model at Kuantan station.

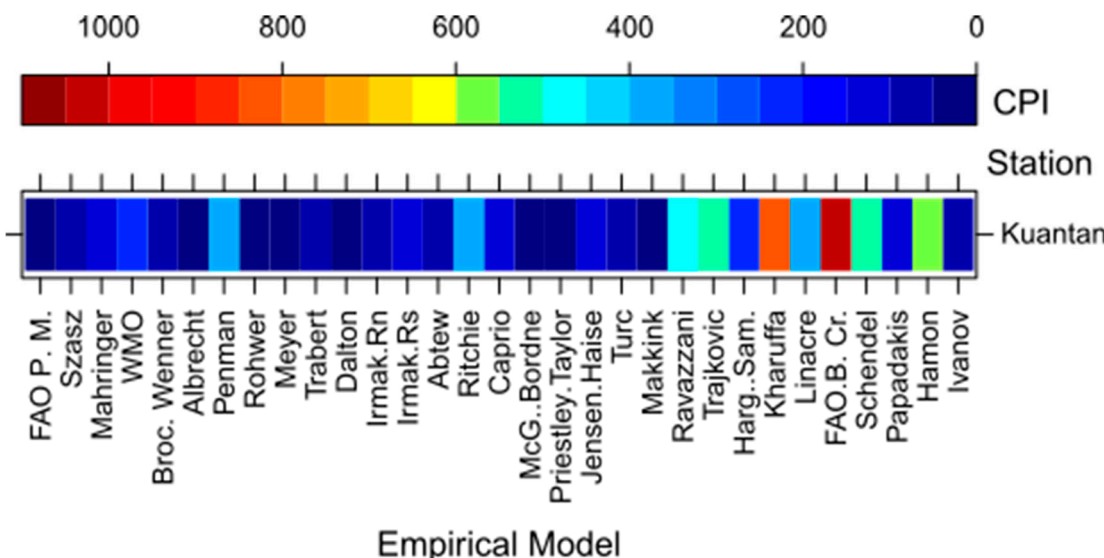

**Figure 4.** The compromise programming index (CPI) of the empirical models at Kuantan station. The FAO Penman-Monteith model had the lowest CPI (7.60), while the FAO Blaney-Criddle model had the highest CPI (1043.44).

*4.3. Ranking the Empirical $ET_o$ Models*

Ranking of the empirical models was done through a six-step procedure, as stated in Section 3.4. First, the CPI values at each station were used to rank the empirical models, where the model that had the lowest CPI was ranked 1st at each station and vice versa. For example, the FAO Penman-Monteith model had the lowest CPI (7.60) in Kuantan station (refer to Figure 4), therefore ranked 1st, followed by the Dalton model which had the 2nd second lowest CPI of 20.08. The rank of each model at each station in Peninsular Malaysia is illustrated in Figure 5 as a level plot.

The frequency of occurrence that a model achieved a certain rank in different stations was calculated. For an example, the FAO Penman-Monteith model was found to have the least CPI in Kuala Terengganu, Kuantan, Melaka, and Muadzam Shah stations, so it was ranked 1st in these stations (refer to Figure 5). Therefore, the frequency of occurrence that the FAO Penman-Monteith model received as number one was four times. The levels of Figure 6 show the complete frequency of occurrence of the empirical models received a certain rank. For example, it can be seen that the FAO Penman-Monteith model was ranked at the 1st rank four times, 2nd rank once, 3rd rank four times,

and 4th rank once. On the contrary, the FAO Blaney-Criddle and Kharuffa models were found to have the highest frequency (10 times) for getting the 31st and 30th rank, respectively.

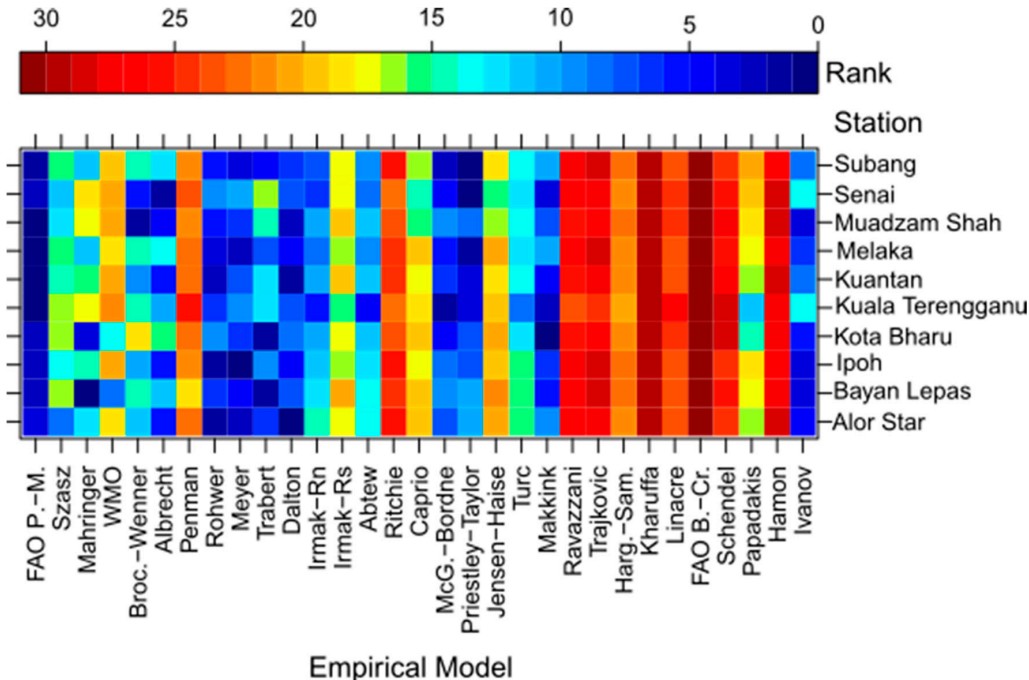

**Figure 5.** Ranking of the empirical models at 10 stations in Peninsular Malaysia according to their CPI (*p* = 1).

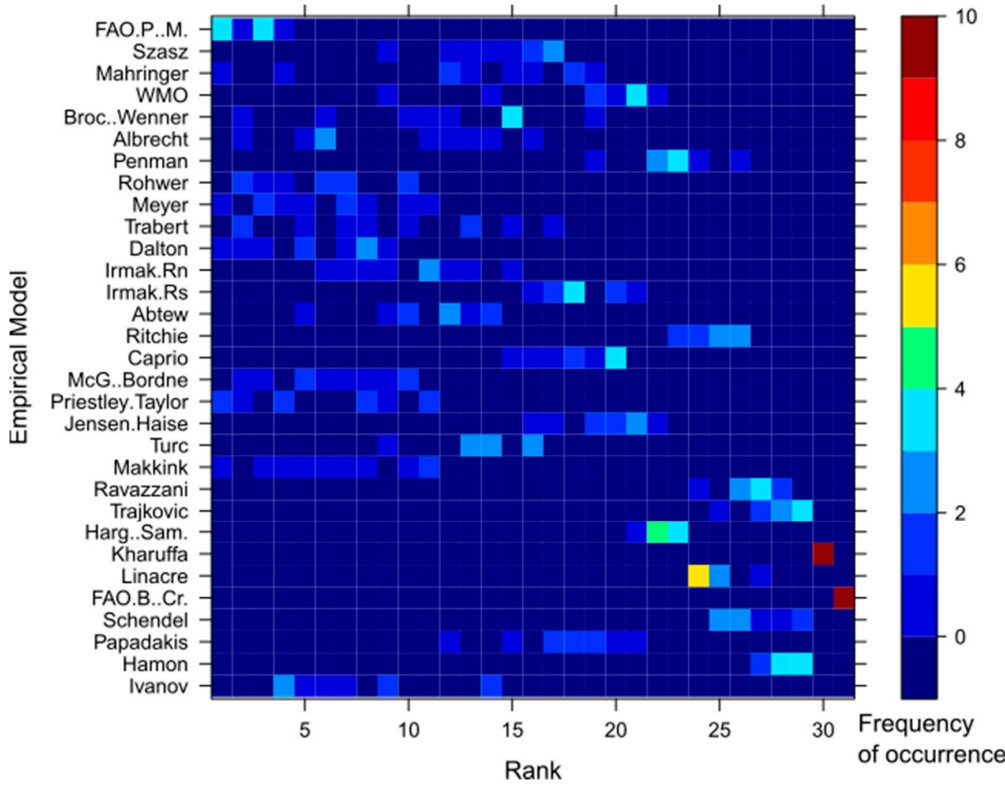

**Figure 6.** The level plot showing the frequency of occurrence of the empirical models for different ranks.

The frequencies of occurrence of rank positions were multiplied by the rank weights and the overall score of each empirical model ($W_m$) was calculated by summing the output of the multiplication.

For example, the frequencies of the FAO Penman-Monteith model having the 1st, 2nd, 3rd, and 4th rank were 4, 1, 4, and 1. So, the $W_m$ was calculated as shown in Equation (8).

$$W_{FAO\ P.-M.} = 4 \times 1^{-1} + 1 \times 2^{-1} + 4 \times 3^{-1} + 1 \times 4^{-1} = 6.08 \tag{8}$$

Based on the $W_m$, the models were finally ranked as shown in Table 4. The FAO Penman-Monteith model was the top-ranked model in this study, followed by the Priestley-Taylor and Dalton models. The Hamon, Kharuffa, and FAO Blaney-Criddle models were ranked as the last three models.

**Table 4.** The overall weight achieved by the empirical $ET_o$ models and their rank for entire Peninsular Malaysia.

| Model | $W_m$ | Final Rank | Model | $W_m$ | Final Rank |
|---|---|---|---|---|---|
| FAO Penman-Monteith | 6.08 | 1 | Papadakis | 0.58 | 17 |
| Priestley-Taylor | 3.54 | 2 | WMO | 0.57 | 18 |
| Dalton | 2.86 | 3 | Caprio | 0.55 | 19 |
| Meyer | 2.72 | 4 | Irmak-Rs | 0.55 | 20 |
| Makkink | 2.50 | 5 | Jensen and Haise | 0.51 | 21 |
| Rohwer | 2.40 | 6 | Hargreaves and Samani | 0.45 | 22 |
| McGuinness and Bordne | 1.98 | 7 | Penman | 0.44 | 23 |
| Trabert | 1.85 | 8 | Linacre | 0.41 | 24 |
| Mahringer | 1.79 | 9 | Ritchie | 0.41 | 25 |
| Ivanov | 1.62 | 10 | Schendel | 0.38 | 26 |
| Albrecht | 1.59 | 11 | Ravazzani | 0.38 | 26 |
| Brockamp and Wenner | 1.26 | 12 | Trajkovic | 0.36 | 28 |
| Irmak-Rn | 1.05 | 13 | Hamon | 0.35 | 29 |
| Abtew | 0.98 | 14 | Kharuffa | 0.33 | 30 |
| Turc | 0.74 | 15 | FAO Blaney-Criddle | 0.32 | 31 |
| Szasz | 0.71 | 16 | - | - | - |

The ranking of the ET models for different values of $p$ is presented in Table 5. The results revealed a slight variation in the ranks of a few models. From example, Makkink was ranked 5th for $p = 1$ and $p = \infty$, and it was ranked 3rd in case of $p = 2$. However, FAO Penman-Monteith was found as the most suitable method for all values of $p$. Priestley-Taylor was found best among the radiation-based models and Ivanov among the temperature-based models for all the cases. However, the best mass transfer-based model was not consistent for all values of $p$. Dalton was found best for $p = 1$, while Meyer was best for $p = 2$ and $p = \infty$. Therefore, both can be considered the most suitable mass transfer-based $ET_o$ models for Peninsular Malaysia.

**Table 5.** Ranking of the empirical $ET_o$ models for different values of $p$ in compromise programming.

| Model | $p = 1$ | $p = 2$ | $p = \infty$ | Model | $p = 1$ | $p = 2$ | $p = \infty$ |
|---|---|---|---|---|---|---|---|
| FAO Penman-Monteith | 1 | 1 | 1 | Papadakis | 17 | 17 | 17 |
| Priestley-Taylor | 2 | 2 | 2 | WMO | 18 | 18 | 18 |
| Dalton | 3 | 5 | 4 | Caprio | 19 | 19 | 19 |
| Meyer | 4 | 4 | 3 | Irmak-Rs | 20 | 20 | 20 |
| Makkink | 5 | 3 | 5 | Jensen and Haise | 21 | 21 | 21 |
| Rohwer | 6 | 6 | 8 | Hargreaves and Samani | 22 | 23 | 22 |
| McGuinness and Bordne | 7 | 9 | 7 | Penman | 23 | 22 | 23 |
| Trabert | 8 | 7 | 6 | Linacre | 24 | 25 | 24 |
| Mahringer | 9 | 8 | 9 | Ritchie | 25 | 24 | 25 |
| Ivanov | 10 | 11 | 12 | Schendel | 26 | 27 | 27 |
| Albrecht | 11 | 10 | 10 | Ravazzani | 26 | 24 | 26 |
| Brockamp and Wenner | 12 | 12 | 11 | Trajkovic | 28 | 28 | 28 |
| Irmak-Rn | 13 | 20 | 15 | Hamon | 29 | 29 | 29 |
| Abtew | 14 | 14 | 13 | Kharuffa | 30 | 30 | 30 |
| Turc | 15 | 15 | 14 | FAO Blaney-Criddle | 31 | 31 | 31 |
| Szasz | 16 | 16 | 15 | - | - | - | - |

To show the efficacy of the top-ranked empirical model identified in this study, the FAO Penman-Monteith estimated and observed $ET_o$ were compared. The heat scatter plots of observed FAO Penman-Monteith $ET_o$ at different stations are presented in Figure 7. The figure shows that most of the points are aligned along the diagonal line, which indicates a perfect estimation of $ET_o$ by FAO Penman-Monteith. The method overestimated $ET_o$ in a few stations, such as Alor Star, Bayan Lepas, Kota Bahru, and Muadzam Shah.

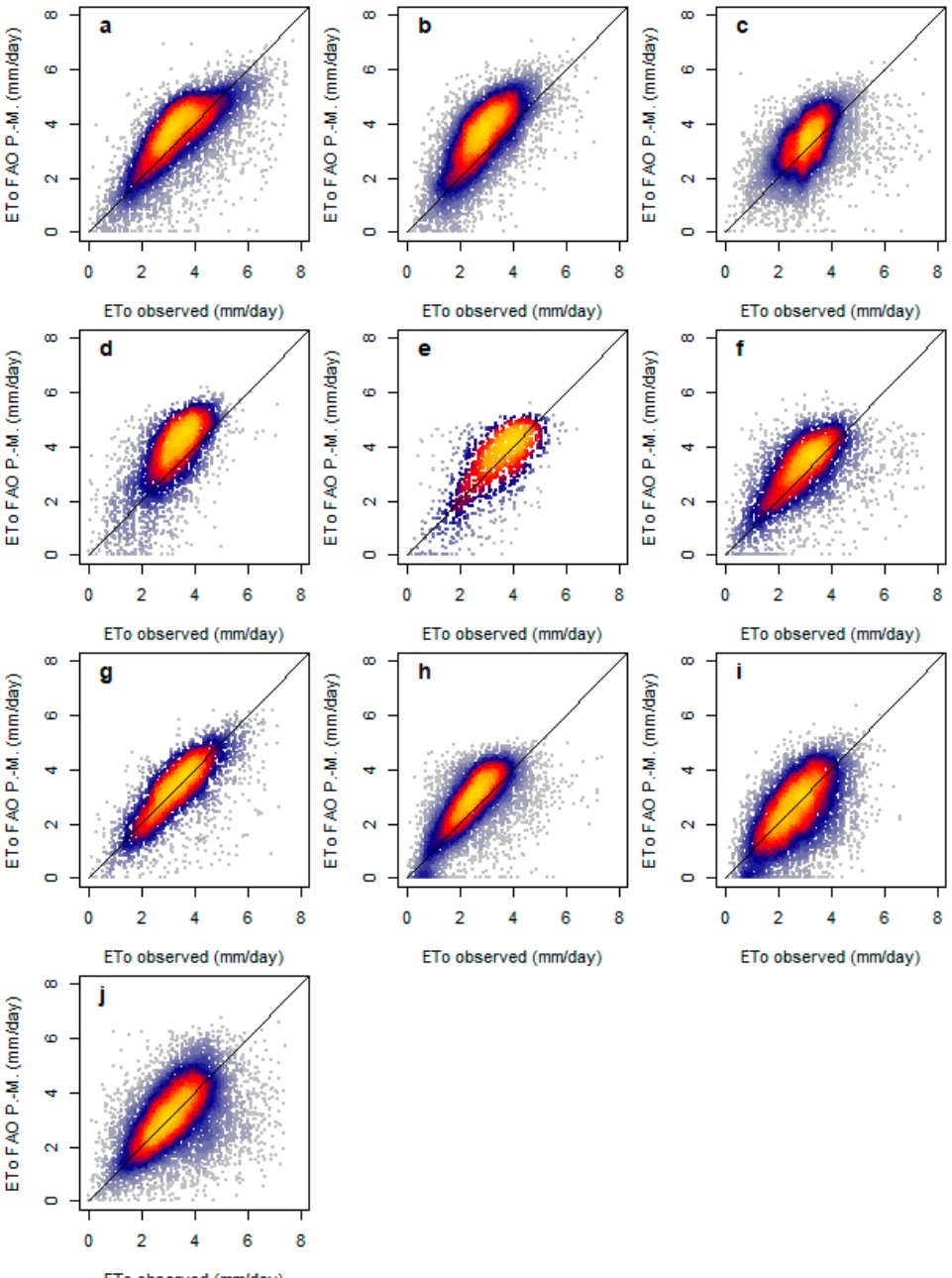

**Figure 7.** Heat-scatter plots of the observed $ET_o$ and the estimated $ET_o$ by the FAO Penman-Monteith model at (**a**) Alor Star; (**b**) Bayan Lepas; (**c**) Ipoh; (**d**) Kota Bharu; (**e**) Kuala Terengganu; (**f**) Kuantan; (**g**) Melaka; (**h**) Muadzam Shah; (**i**) Senai; and (**j**) Subang.

## 5. Discussion

The FAO–Penman-Monteith method has been found as the most efficient for the estimation of $ET_o$ in different climatic regions. The FAO Penman-Monteith model was developed based on physiological

and aerodynamic theories of surface water release to the atmosphere. Therefore, it can be used as a standard model for estimation of $ET_o$ in any region without any adjustment of parameters [24] based on the FAO recommendation [13,15]. The present study also found that FAO Penman-Monteith is the best model for Peninsular Malaysia. However, the Penman-Monteith model needs a large number of meteorological variables, including air temperature, wind speed, relative humidity, and solar radiation, for the estimation of $ET_o$. It is often very difficult to obtain data of all the meteorological variables. Many meteorological stations in the developing country do not measure all these variables. Therefore, a large number of alternative models have been developed based on the availability of data. The success of those models in a particular area often depends on the climate of the region. Therefore, the selection of an appropriate model based on the availability of data and the performance of $ET_o$ estimation model is a difficult task. The performance of 31 empirical $ET_o$ models has been assessed in this study. Input requirements of the models are different.

The performance of empirical $ET_o$ models was often found to vary from station to station within the same climate zone, which may be due to the period and quality of data used, and uncertainty in the coefficient values used for the estimation of $ET_o$. Besides, suggesting different models for different stations often makes the practical application of the $ET_o$ estimation model complex. Therefore, a single model is often suggested for the regional level for the estimation of $ET_o$. Thus, the ranking of $ET_o$ estimation models in different stations was used in this study for the ranking of $ET_o$ models for the entire Peninsular Malaysia using information aggregation approach.

In the present study, radiation-based Priestley and Taylor was found to perform best after the FAO Penman-Monteith model. It is followed by the mass transfer-based Dalton and Meyer models. The Priestley and Taylor model needs three meteorological variables (mean air temperature, solar radiation, and relative humidity) compared to the five variables required by the Penman-Monteith model (temperature, solar radiation, relative humidity, wind speed, and saturated vapor pressure), while Dalton and Meyer need three meteorological variables (mean temperature, relative humidity, and wind speed). Based on the availability of meteorological data, an appropriate model can be selected for the estimation of $ET_o$ in Peninsular Malaysia with more or less similar accuracy.

Among the temperature-based models, only the Ivanov model was found to perform satisfactorily, which was ranked 10th among the 31 models compared in the present study. Other temperature-based models performed the worst and were ranked at the bottom of all the models. Many of the models were developed for a particular climate. For example, the Priestley and Taylor and the Makkink models were developed for the estimation of $ET_o$ in a humid climate. On the other hand, the Turc model was found suitable for $ET_o$ estimation in a cold, humid and arid climate [26]. Therefore, the Priestley and Taylor and Makkink models were found to perform very well among the radiation-based models, while Turc was found to perform worse than the simple temperature-based Ivonov model in tropical Malaysia.

The findings of the present study contradicts earlier studies. Ali and Lee [31] found Blaney–Criddle as the most suitable model after Penman-Monteith for the estimation of $ET_o$ at Alor Setar station in Peninsular Malaysia. They only used relative error for the assessment of the performance of empirical $ET_o$ models. Tukimat et al. [13] assessed the performance of seven empirical ET models for the same station using three statistical metrics, namely absolute error, relative error, and correlation coefficient. They found the least absolute and relative errors for the Hargreaves-Samani but highest correlation for Makkink, followed by Priestley-Taylor and Turc. They come to an overall conclusion that radiation-based models are most suitable for the estimation of $ET_o$ in the region, which also support the findings of the present study. But they failed to decide the best model due to a contradiction in statistical metrics. Lee et al. [29] compared the performance of eight empirical models using mean absolute error, and reported FAO Blaney-Criddle as the most suitable model after FAO Penman-Monteith for estimation of $ET_o$ in the west coast of the peninsula. Muniandy et al. [32] compared the performance of 26 empirical models at Senai station using eight statistical metrics. They also obtained contradictory results in term of different statistics. They took the arithmetic mean of the statistics to rank the models and found Penman as the best among the mass transfer-based models, McGuinness and Bordne among

the radiation-based, and Szasz among the temperature-based models. Different models have been reported as the best in different stations in Peninsular Malaysia in the above studies, which do not match with the findings of the present study. This is due to the use of a single statistic for making a decision, as in the studies of Lee et al. [29] and Ali et al. [31]. Tukimat et al. [13] and Muniandy et al. [32] used multiple statistics, but did not attempt to find the best ET model based on the statistics. Muniandy et al. [32] attempted to rank the models based on the average of multiple statistics, but the average of statistics does not provide an optimum solution as the ranges of statistics metrics vary widely.

CP was used in this study for finding the most suitable empirical models based on four statistical metrics which can be used to measure the similarity between two time series in a robust way. CP proves a robust model compared to many MCDA models for finding a reliable solution based on multiple contradictory objectives. Therefore, the best empirical models identified in this study based on CP can be considered more reliable. Besides, the empirical models were ranked for the entire Peninsular Malaysia considering the fact of the same tropical humid climate for the whole region. The information aggregation model was used in this study for this purpose, which ranks the models based on the frequency of rank obtained by different models in different stations. Therefore, the top-ranked models in different stations were also found to achieve the top rank for the entire peninsula. This indicates the ranking of the models obtained in this study for the entire peninsula can be used for finding the most suitable model based on the availability of data for reliable estimation of $ET_o$.

## 6. Conclusions

The CP and GDM methods were used in this study for the ranking 31 $ET_o$ empirical models for the estimation of $ET_o$ in Peninsular Malaysia, based on four statistical metrics applied at 10 locations distributed over the study area. The result revealed Priestley and Taylor as the most suitable among the radiation-based models, Dalton among the mass transfer-based models, and Ivonov among the temperature-based models for the region. Though the mass transfer-based models were found more reliable compared to radiation-based models, Priestley and Taylor was found as the most suitable after Penman-Monteith, which is globally considered as the standard model for $ET_o$ estimation. The Priestley and Taylor model needs only mean air temperature, solar radiation, and relative humidity compared to a large number of meteorological variables required for the estimation of $ET_o$ using Penman-Monteith. Therefore, Priestley and Taylor can be used as a replacement of Penman-Monteith in the estimation of $ET_o$ when available data is limited. The present study suggests that the Ivonov model, which requires only mean temperature and relative humidity, can be used for the worst case in terms of availability of data.

Estimation of $ET_o$ in this study was based on pan coefficient of 0.75, as suggested by the Department of Irrigation and Drainage of Malaysia. The sensitivity of the ranking of $ET_o$ estimation methods can be tested in the future for different pan coefficients. CP and GDM were used in this study for making a decision on $ET_o$ models. Beside CP and GDM, other decision-making and information aggregation methods can be used, and their performance can be compared with the findings of the present study in the future. The parameters of the empirical models can be calibrated for Peninsular Malaysia before the comparison and ranking of the models.

**Author Contributions:** M.K.I.M., S.S., and T.b.I. conceived and designed this study; M.K.I.M., M.S.N., Y.H.S., and E.-S.C. analyzed the data; S.S. and M.S.N. wrote the first draft; M.S.N. prepared the figures; S.S., E.-S.C., and T.b.I. wrote the final manuscript.

**Funding:** This work was supported by the Korea Environmental Industry & Technology Institute (KEITI) grant funded by the Ministry of Environment (Grant 19AWMP-B083066-06).

**Acknowledgments:** The authors would like to thank the reviewers for their valuable comments which improved the quality of this paper.

**Conflicts of Interest:** The authors declare no conflict of interest. The funders had no role in the design of the study; in the collection, analyses, or interpretation of data; in the writing of the manuscript, or in the decision to publish the results.

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
