# Peer review of "Evaluation of Empirical Reference Evapotranspiration Models Using Compromise Programming: A Case Study of Peninsular Malaysia"

_sustainability, doi:10.3390/su11164267_

Round 1
Reviewer 1 Report
The authors present a manuscript titled “Evaluation of 31 Empirical Reference Evapotranspiration Models Using Compromise Programming”. The manuscript is of very good quality. Some improvements require. In methods and data, the observed ETo was calculated by multiplying the pan evaporation data by the pan coefficient of 0.75. Generally, there is a wide range of pan coefficient values. Caution should be used with any pan coefficient method for prediction of ET, the method is weak to begin with and coefficients will experience a strong seasonal variation that follows vegetative growth patterns. Please clarify use pan coefficient of 0.75 related to type of land cover in Malaysia. In the conclusion, please refer to the limitation of using the pan coefficient.
Author Response
The authors present a manuscript titled “Evaluation of 31 Empirical Reference Evapotranspiration Models Using Compromise Programming”. The manuscript is of very good quality. Some improvements require. In methods and data, the observed ETo was calculated by multiplying the pan evaporation data by the pan coefficient of 0.75. Generally, there is a wide range of pan coefficient values. Caution should be used with any pan coefficient method for prediction of ET, the method is weak to begin with and coefficients will experience a strong seasonal variation that follows vegetative growth patterns. Please clarify use pan coefficient of 0.75 related to type of land cover in Malaysia. In the conclusion, please refer to the limitation of using the pan coefficient.
Answer: Thank you for your comment. The issue has been discussed and the justification of the selection of pan coefficient of 0.75 has been given in the revised manuscript as below:
“Pan evaporation is an indirect and less expensive method of estimation of ET and therefore, it is most widely used for estimation of ET. The pan evaporation data is multiplied by the pan coefficient to get the ETo. The pan coefficient value varies between 0.35 and 0.85 depending on the nature of the evaporating surfaces (land use), altitude, average humidity and average wind speed of the site [47]. Considering existing setup, by Department of Irrigation and Drainage of Malaysia [48] suggested a pan coefficient of 0.75 for the estimation of ETo from pan evaporation in Malaysia. Therefore, the observed ETo was calculated by multiplying the pan evaporation data by the pan coefficient of 0.75.”
Besides, it is mentioned as limitation in conclusion as suggested by reviewer. Following texts have been added in conclusion for this purpose:
“Estimation of ETo in this study was based on pan coefficient of 0.75 suggested by Department of Irrigation and Drainage of Malaysia. The sensitivity of the ranking of ETo estimation methods can be tested in future for different pan coefficients.”
Reviewer 2 Report
With your manuscript you provided a very concise, convincing paper, giving a diligently performed and comprehensive inspection of the full set of evapotranspiration models. There are but a few questions and remarks which I denoted in the annotated text file.

Author Response
With your manuscript you provided a very concise, convincing paper, giving a diligently performed and comprehensive inspection of the full set of evapotranspiration models. There are but a few questions and remarks which I denoted in the annotated text file.
Answer:
Thank you for your encouraging comments. We revised the paper based on your comments. All of the suggestions have been considered during revision of the article. Details of the corrections made are mentioned in point-to-point answer to your comments provided on the PDF version of the paper. The answers are as follows:
Comment:
L32: Evaporation accounts for a real deficiency, while the transpiration component constitutes a 'productive loss'. Thus, a more impartial concept for the combined evaporation + transpiration would be 'water release' (cf. L51/52).
Answer:
Thank you for your comment. We agree to your comment. Evapotranspiration is mentioned as “water release to the atmosphere” instead of “water loss” throughout the manuscript.
Comment:
L67: Of course, you should mention the primary role of rainfall...
Answer:
Thank you for your comment. It is correct, rainfall plays the primary role of high biodiversity in the region. Besides rainfall, ET and other factors also play important role. Therefore, the sentence is revised as below:
“This rich biodiversity is promoted by high rainfall and high ET among the other factors.”
Comment:
L74:76: et al.
Answer: “, et al”. was added to the text.
Comment:
L135: Small values because of permanently high air humidity?
Answer:
Yes, you are right. The sentence is revised as below:
“Furthermore, it is lower in central mountainous areas (2.5 mm/day) due to relatively higher humidity compared to the coastal region (4 – 5 mm/day) where the humidity is less.”
Comment:
Table 1, header: mm/day
Answer: The unit of ET was corrected in the Table 1 header to “mm/day”.
Comment:
L318: scatter plots
Answer:
The sentence was corrected based on this comment as follows.
“The heat scatter plots of observed and FAO Penman-Monteith ETo at different stations are presented in Figure 7.”
Comment:
L332: cf. comment at L32
Answer:
Evapotranspiration is mentioned as “water release to the atmosphere” instead of “water loss”
Comment:
L334: is
Answer: The sentence was corrected based on this comment as follows.
“The present study also found that FAO Penman-Monteith is the best model for Peninsular Malaysia.”
Comment:
L358: is followed by
Answer: The sentence was corrected based on this comments as follows.
“It is followed by the mass transfer-based Dalton and Meyer models.”
Comment:
L384: Tukimat et al.
L400:402: write 'et al.'
Answer: Thank you. “et al.” was used. The reference style was revised and the citation in the whole manuscript was corrected for similar issues.
Comment:
L411: is/was
Answer: The sentence was corrected as follows.
“The information aggregation model was used in this study for this purpose, which ranks the models based on the frequency of rank obtained by different models in different stations.”
Reviewer 3 Report
1. The manuscript needs a full English review. I have specified some parts of the text where English should be corrected. But, an English specialist is required to improve the whole document.
2. There is a major concern in the use of Compromise Programming (CP). The authors confuse its applicability, mistakenly opening the possibility that CP is able to find optimal solutions. CP does not provide optimal solutions. Compromise programming is only a technique for estimating the minimum distance between the efficient frontier and the ideal point based on the level of importance assigned to each criterion. Besides, authors use only one p value among the three that are typically referenced. The use of only one p value is not recommended for CP applications due to the uncertainty of the researcher’s preferences. The distance for any value of p (where p varies between 1 and infinity) is used here as a measure to express individual preferences and represents the risk in decision-making, comparisons, similarities, or proximity of individual coordinates (Perez-Verdin, et. al. 2018, Forests).
3. The use of these p values allows the researcher to perform a sensitivity analysis and test the robustness of the solutions. This sensitivity test is not provided in the manuscript.
4. To overcome this issue, authors need to estimate CP values, in addition to the already performed (p=1), for p=2 and p=infinity and perform a sensitivity analysis. Although it sounds overwhelming, this can easily be done in a spreadsheet. Once CP calculations are done for the three p values, the authors can see if solutions are consistent. If selection of the most preferred model is inconsistent, then they need to decide which p value would be best to represent their preferences.
5. In addition, the CP estimation does not consider the relative levels of importance (or weights) for each criterion. While this may be optional, sometimes it helps to recognize which model, for example, has the highest acceptation by experts or is more recommended in the literature. Authors need to explain why they are not considering this step in CP estimations.
Other things:
6. Parts of the methods and results are repeated in the discussion. This section should be revised and improved.
7. There are other comments made in the pdf manuscript

Author Response
Comment:
1. The manuscript needs a full English review. I have specified some parts of the text where English should be corrected. But, an English specialist is required to improve the whole document.
Answer:
Thank you for your comment. English language of the paper is carefully checked by an experienced scientific article writer. All the grammatical mistakes have been corrected.
Comment:
2. There is a major concern in the use of Compromise Programming (CP). The authors confuse its applicability, mistakenly opening the possibility that CP is able to find optimal solutions. CP does not provide optimal solutions. Compromise programming is only a technique for estimating the minimum distance between the efficient frontier and the ideal point based on the level of importance assigned to each criterion. Besides, authors use only one p value among the three that are typically referenced. The use of only one p value is not recommended for CP applications due to the uncertainty of the researcher’s preferences. The distance for any value of p (where p varies between 1 and infinity) is used here as a measure to express individual preferences and represents the risk in decision-making, comparisons, similarities, or proximity of individual coordinates (Perez-Verdin, et. al. 2018, Forests).
Answer: Thank you for your comments. We revised the articles based on your suggestion. We used different values of p to estimate the ranking of ET models. Details are mentioned in method section. Obtained results are presented and discussed in the revised version of the article. A new table (Table 5) has been added for this purpose.
Comment:
3. The use of these p values allows the researcher to perform a sensitivity analysis and test the robustness of the solutions. This sensitivity test is not provided in the manuscript.
Answer: Thank you for your comments. We assessed the sensitivity of the ranking using different p values following your suggestion. Obtained results are presented in Table 5. The results are discussed as below:
“The ranking of the ET models for different values of p is presented in Table 5. The results revealed a slight variation in the ranks of few models. From example, Makkink was ranked 5 for p=1 and p=¥, it was ranked 3 in case of p=2. However, FAO Penman-Monteith was found as the most suitable method for all values of p. The Priestley-Taylor was found best among the radiation based models and Ivanov among the temperature-based models for all the cases. However, the best mass-transfer based model was not consistent for all values of p. Dalton was found best for p=1, while Meyer was best for p=2 and ¥. Therefore, both can be considered most suitable mass-transfer based ETo models for the peninsular Malaysia.”
Comment:
4. To overcome this issue, authors need to estimate CP values, in addition to the already performed (p=1), for p=2 and p=infinity and perform a sensitivity analysis. Although it sounds overwhelming, this can easily be done in a spreadsheet. Once CP calculations are done for the three p values, the authors can see if solutions are consistent. If selection of the most preferred model is inconsistent, then they need to decide which p value would be best to represent their preferences.
Answer: Thank you for your comments. We assessed the sensitivity of the ranking using different p values following your suggestion. Best models were found consistent for most of the cases. Obtained results are described in revised manuscript with a newly added tables. Details are also given in answer to your previous comment.
5. In addition, the CP estimation does not consider the relative levels of importance (or weights) for each criterion. While this may be optional, sometimes it helps to recognize which model, for example, has the highest acceptation by experts or is more recommended in the literature. Authors need to explain why they are not considering this step in CP estimations.
Answer:
Thank you for your comment. No statistical metric is superior to another one. Those are always considered to have equal importance. Therefore, we also considered equal importance and weight parameter is not used. This is mentioned in method as below:
” In this study, we considered equal importance of all the statistical metrics used to measure the performance of ETo estimation models and therefore, weight parameter of CP method proposed by Zeleny [36] is not considered.”
Other things:
6. Parts of the methods and results are repeated in the discussion. This section should be revised and improved.
7. There are other comments made in the pdf manuscript
The comments of the PDF file are:
Title: 31 I suggest to shorten the title.
Answer: The number 31 was deleted for the title.
L16: You say nothing (other that it is important). The next sentence says the same
Answer: This sentence was deleted and merged with the 2nd sentence based on this comment. The 1st sentence is now as follows:
“Selection of an appropriate empirical reference evapotranspiration (ETo) estimation models is very important for the management of agriculture, water resources, and environment.”
L22: study area....We don´t know what do you mean by a region.
Answer: The abstract was revised. The word region was replaced by “a given study area”. The sentence is now as follows.
“Besides, ranking of ETo estimation methods for a given study area based on the rank at different stations is also a difficult task.”
L25: “ ETo in Peninsular Malaysia”
Answer: “Peninsular Malaysia” was removed.
L32: Don't just say things are important (everything is important). Go the 'important' point!. Start the sentence like this: Evapotranspiration, the process of water loss to the atmosphere, plays a crucial role in irrigation...
Answer: The sentence was rewritten based on this comment as follows.
“Evapotranspiration (ET), the process of water loss to the atmosphere, plays a crucial role in irrigation management [1], water balance estimation [2], surface runoff estimation [3], groundwater level prediction [4], water stress assessment [5], reservoir management [6], daily flux modelling [7], and climate change impact assessment [8].”
L49: Owing to the (because of)
Answer: “owing to” was replaced by “because of”
L55: (iv) combination. (a combination of the above)
Answer: The sentence was corrected to be as follows.
“Most widely, it is classified into four groups: (i) water balance/mass transfer, (ii) radiation, (iii) temperature, and (iv) combination of the aforementioned”
L67: among other factors
Answer: The sentence was rewritten based on this comment as follows.
“This rich biodiversity is promoted by the high ET among other factors.”
L70: the
Answer: “the” was added to the text as follow.
“A large number of studies have been conducted to select the suitable ETo model in different parts of the globe [15,20,24-27] and in Peninsular Malaysia [13,28-31].”
L93: needs citation
Answer: Citation of papers having the problem of a statistical metrics contradictory results were added.
L96: CP does not provide optimal solutions. As a multcriteria decision method, it does offer the most suitable option
Answer:
Thank you for your comment. Corresponding texts are revised. The term “optimal solution” is replaced with “the most suitable solution”.
L98: No... please revise this statement. there are not optimal values. Optimal values can only be obtained when restrictions are imposed and met. The linear programming technique does provide optimal solutions as long as restrictions are met.
Answer:
The sentence is revised as:
“The CP attempts to identify a solution where all the considered objectives achieve the most suitable value.”
L110: of
Answer: The sentence was corrected as follows.
“The objective of the present study is to use CP for the ranking of empirical ETo models for Peninsular Malaysia.”
L120: The map provides that info
Answer: The latitude and longitude were removed from the text.
L135: the and hour and high in the dry 135 season when the sunshine hour is longer.
Answer: “The” was added to the sentence as follows.
“The ETo is low in the rainy season due to less sunshine hour.”
L138: Suggest to get help with English use. An English specialist can refine some minor grammatical errors.
Answer: The sentence was corrected as follows.
“Furthermore, it is lower in central mountainous areas (2.5 mm/day) due to relatively the lower temperature compared to the coastal region (4 – 5 mm/day) where the temperature is higher.”
L142: of information
Answer: the heading was rewritten based on this comment as follows.
“2.2. Data and Sources of information”.
L146: shown
Answer: “given” was replaced by “shown” as follows.
“The locations of the meteorological stations are shown in Figure 1.”
L148: Descriptive statistics of the meteorological stations...
Answer: The Table 1 captain was rewritten based on this comment as follows.
“Table 1. Descriptive statistics of the meteorological stations used in the present study.”
L153: performance
Answer: “Skill” was replaced by “performance” based on this comment as follows.
“The performance of different empirical ETo models was assessed and ranked by comparing their estimations with the in-situ data”
L154:159: Delete the sentences used to described the steps used in the study.
Answer:
Thank you for your suggestion. We mentioned the steps used in the study for easy understanding of the methodology used for the ranking of ET methods. Therefore, we kept it.
L161: Estimation of empirical ET models
Answer: Thank you for your suggestion. We described the empirical models used in the study for the estimation of ET. Therefore, we kept the heading of the section as “Empirical ETo models”
L167: location using the models
Answer: This part was deleted. The revised sentence is as follows.
“The ETo was calculated using the meteorological input at each station location without any calibration.”
L192: CAPS
Answer: “md” is the common term when referring to the modified index of agreement. Therefore, “md” was kept as it is in the whole manuscript.
L207: as in equitation 5. (as follows)
Answer: “as follows” was used in the text.
L207: I don't see the weight parameter, which is implicit in CP use. This weight can be given by the researcher or expert consultation
Answer: Thank you for your observation. We considered equal weight. Therefore, no weight parameter is given. This is mentioned in method as below:
” In this study, we considered equal importance of all the statistical metrics used to measure the performance of ETo estimation models and therefore, weight parameter of CP method proposed by Zeleny [36] is not considered.”
L210: It can also be infinite
Answer: The sentence was rewritten based on this comment as follows.
“The parameter p value can be either 1, 2 or infinite to measure the linear or squared Euclidean distance.”
L211: Again, CP cannot provide optimal solutions. I suggest to read Zeleny for details.
Answer:
We revised the sentence as below:
“The CPI values ranges between zero and positive infinity where zero is the best value.”
L213:218: check verb tense for your explanations. Overall, The past tense is used to describe things that have already happened
Answer:
Thank you for your comments. Those lines are used to justify the ranking method used. Generally practiced procedure is described. Therefore, present tense is used.
L213: the metrological variables.
Answer: Sorry for this silly mistake. The corrected sentence is as follows.
“The ETo was estimated using all the 31 empirical models at each station using the meteorological variables.”
L234: Can you explain, why are these overestimations?
Answer:
The sentence is modified. The possible cause is mentioned as below:
“which indicates that the overestimation may be due to non-consideration of other factors influencing ETo in the study area.”
L238: suggest adding "based on the metrics, the mass transfer-based models....Or saying that "much perfect" is based on what?
Answer: Till this point in the manuscript the metrics results were not shown. The sentence was rewritten as follows.
“Overall, the mass transfer-based models’ estimations, Figure 2 (u-ad), were found more aligned to the 1:1 diagonal line than the temperature- and radiation-based methods.”
L286: Is there a citation or a previously published work?
Answer: The sentence was rewritten to be clearer as follows.
“Ranking the empirical model was done through a six-step procedure as stated in section 3.4.”
L288:289: an…. and…..top (first)
Answer: the text was corrected based on these comments.
“For example, the FAO Penman-Monteith model had the lowest CPI (7.60) in Kuantan station (refer to Figure 4), therefore ranked first followed by Dalton model which had the 2nd lowest CPI of 20.08.”
L293: Ranking
Answer: “Ranking” was used instead of “the Rank” in the figure caption.
L299: as number one, was four times
Answer: The sentence was rewritten based on this suggestion.
“Therefore, the frequency of occurrence that the FAO Penman-Monteith model received as number one, was four times. The levels of Figure 6 show the complete frequency of occurrence of the empirical models received a certain rank.”
L301: needs English review
Answer: English was revised by professional academic review. The corrected sentence is as follows. “For example, it can be seen that the FAO Penman-Monteith model was ranked at the 1st rank: 4 times, 2nd rank: once, 3rd rank: 4 times and 4th rank: once.”
L321: at in a
Answer: The “in a” was used instead of “at”. The sentence after correction is as follows.
“The method overestimated ETo in a few stations such as Alor Star, Bayan Lepas, Kota Bahru, and Muadzam Shah.”
L322: However, still, the majority of the 322 points were found aligned to the diagonal line.
Answer: This sentence was deleted in the revised manuscript based on this comment.
L329: Actually, it is your recommendation (based on your results)... What is the World Food Organization??? Are you talking about FAO?
Answer: The sentence was omitted in the revised manuscript.
L342:348: You already said this in methods
Answer:
Thank you for your observation. Corresponding sentences are removed.
L369:377: You are just repeating your results...
Answer:
Thank you for your observation. Corresponding sentences are removed.
L381: contradicts with the previous studies. (Other)
Answer: The word “earlier” was used instead of “previous”.
L382: They simply used (Only).
Answer: “only” replaced “simply” in the revised manuscript as follows.
“They only used relative error for the assessment of the performance of empirical ETo models.”
L400: what happened here?
Answer: The sentence was corrected as follows.
“This is due to the use of a single statistics for making a decision as in the studies of Lee, et al. [28] and Ali, et al. [30].”
L407: Your results lack of important procedures:
1. You are not using the weights parameter as Zeleny suggests
2. You are only using one p value
3. Because of that, there is not sensitivity analysis to check the robustness of your solutions
Answer:
As we mentioned earlier, we considered equal importance and therefore, weight parameter is not used. This is mentioned in method as below:
” In this study, we considered equal importance of all the statistical metrics used to measure the performance of ETo estimation models and therefore, weight parameter of CP method proposed by Zeleny [36] is not considered.”
Round 2
Reviewer 3 Report
Some comments I made in my first review seem to be not addressed. Please provide a rebuttal of any discrepancy you may have with my comments. I indicated a few of them in your revised version (see pdf document). Also, be more explicit in the meanings of the values of p in the CP calculations. I recommend to visit any Zeleny’s work for further clarifications.
